# Effects of Nanobubbles in Dermal Delivery of Drugs and Cosmetics

**DOI:** 10.3390/nano12193286

**Published:** 2022-09-21

**Authors:** Yuri Park, Soyeon Shin, Nutan Shukla, Kibeom Kim, Myoung-Hwan Park

**Affiliations:** 1Department of Convergence Science, Sahmyook University, Seoul 01795, Korea; 2Convergence Research Center, Nanobiomaterials Institute, Sahmyook University, Seoul 01795, Korea; 3Department of Chemistry and Life Science, Sahmyook University, Seoul 01795, Korea; 4N to B Co., Ltd., Business Incubator Center #5002, Hwarang-ro, Nowon-gu, Seoul 01795, Korea

**Keywords:** nanobubbles, dermal delivery, cosmetics, depigmentation

## Abstract

Dermal delivery, which delivers drugs and cosmetics through the skin, has attracted significant attention due to its non-invasive and simple administration compared with oral or injectable administration. However, delivery of the ingredients through the skin barrier is difficult because the primary function of the skin is to protect the human body by preventing the invasion of contaminants. Although various techniques have been developed to overcome skin barriers, chemical toxicity, complicated processes, and expensive equipment still remain as obstacles. Moreover, green chemistry, which minimizes or eliminates the use of toxic chemicals, is required in the cosmetic industry. Thus, the development of a new method for dermal delivery is required. In this study, we provide a new method for dermal delivery using nanobubbles (NBs). NBs generated in oil improve the delivery effect of the active ingredients through the high Brownian motion and charge-balancing effect. Franz cell experiments and depigmentation experiments using the B16F10 melanoma cells were conducted to confirm the enhanced delivery effects. The system using NBs will contribute to the advancement of the dermal delivery of drugs and cosmetics.

## 1. Introduction

For thousands of years, people have applied various substances to the skin to achieve therapeutic effects [1]. Dermal delivery has received considerable attention in recent years in the fields of medicine and cosmetics because it can be a non-invasive alternative to conventional oral or injectable administration using the skin, which is the largest organ in the human body [1,2,3]. Indeed, the delivery of ingredients through the skin offers a variety of advantages, including the improved bioavailability of ingredients that suffer from the gastrointestinal environment or hepatic first-pass effects, the possibility of long-term drug delivery at a constant rate, reduced side effects, and improved patient compliance [4,5,6,7]. For cosmetics used in daily life to improve the skin condition, a non-invasive and self-administered method through the skin is preferred over a delivery method that is administered by professionals and accompanied by pain. However, the skin is the first physiological barrier that protects human and mammalian bodies against the invasion of pathogens and chemical/physical assaults [8,9]. This barrier function of the skin, which is a significant obstacle in dermal delivery, limits the delivery amount of the active ingredients [9,10]. Thus, the development of formulations (gels, hydrogels, organogels, nano emulsion, micelle, liposome, and exosome), or techniques using physical force (electrical voltage, ultrasound, micro-needles, thermal ablation, magnetophoresis, photomechanical waves, and electron beam irradiation) are being continuously sought to overcome these limitations to enhance the delivery effect [11,12,13,14,15]. Nevertheless, chemicals commonly used in formulations are biologically toxic and can cause skin irritation, and the equipment required for the technology using physical force is large and expensive [16]. Moreover, in the cosmetic industry, green chemistry is preferred or required, and the use of harmful chemicals should be minimized. To overcome these obstacles, new dermal delivery systems should be developed with minimal or no use of chemicals and equipment.

Nanobubbles (NBs), defined as gas cavities with a submicron scale, have drawn increased attention in the field of drug delivery systems as a vehicle due to their several novel properties [17,18,19]. First, NBs are invisible to the naked eye because their diameter is smaller than the wavelength of visible light, and a NB solution is completely transparent [20,21]. Second, NBs are highly stable in solution for several months because of stronger Brownian motion than the buoyancy [22,23,24]. Third, NBs, which have a high surface area, can load the ingredients between the NBs and the solvent interface [25,26,27]. Finally, a burst of NBs via external stimuli such as sonication and physical force generates energy that enhances the delivery of the ingredients by increasing the diffusion effect [28,29,30,31,32]. NBs with these unique characteristics are used for the delivery of the active ingredients in various fields, and they have great advantages, especially in the cosmetic industry [33,34,35,36]. Since they are invisible in the solution, they can be applied to various cosmetics products, and their long-term stability ensures that the efficacy of functional cosmetics will be maintained over time. Although NBs have an attractive advantage in the field of dermal delivery and the cosmetics industry, their effects on improving the delivery of ingredients without specific stimulus have not been reported yet on the dermal delivery system. 

Here, we report a new dermal delivery technique using NBs contained in oil to improve the delivery effect of active ingredients. The NB system is prepared by the simple mixing of the NBs and the active ingredients, and it does not require additional chemical or complicated experimental procedures. The high Brownian motion of the NBs is expected to increase the diffusion of the active ingredient, and the charge-balancing effect of NBs is expected to improve the movements of the active ingredient in the skin barrier. NBs were generated using a novel, previously patented device, and Nile red coloring was added to the NB oil for the additional experiments. [37,38] After the preparation of the solution, the delivery effect of the active ingredients was confirmed using the Franz cell method, which is proposed as a reliable alternative to animal testing for skin absorption testing [39]. Franz cells are divided into two compartments with an artificial membrane mimicking the skin, and the active ingredients were delivered through the membrane [40,41,42,43,44]. Finally, the improved effect of the active ingredients by increasing the delivery effect was confirmed through an in vitro experiment. Delivery test was further conducted using α-bisabolol, which is a representative depigmenting ingredient [45,46,47]. Aliquots were collected from the delivery experiment and then were incubated with the B16F10 melanoma cells to confirm the depigment effect.

## 2. Materials and Methods

The Material: Medium chain triglyceride (MCT) was supplied from New&New Co., LTD. (Chungnam, Korea). Oxygen was purchased from Daehan Chemical Co., Ltd. Nile red was purchased from the TCI (Tokyo, Japan). Mixed Cellulose Ester (MCE) membrane was purchased from Scilab Korea Co., LTD. (Seoul, Korea). Nanoparticle tracking analysis (NTA; NanoSight NS300, Malvern Instruments, Worcestershire, UK,). The Franz cell, which was produced on a custom order from Dail Scientific Trade, Inc. (Seoul, Korea) was used to analyze the delivery effect of active ingredients. Microplate reader (Tecan Trading AG, Zurich, Switzerland,) was used to measure to Nile red and melanin concentration. B16F10 melanoma cells were obtained from the Korean Cell Line Bank (Seoul, Korea). All cell reagents for in vitro studies such as phosphate-buffered saline solution, Dulbecco’s modified Eagle’s medium (DMEM), fetal bovine serum (FBS), and penicillin–streptomycin were all purchased from Sigma–Aldrich (St. Louis, MO, USA).

Generation of NBs in oil: NBs were produced by placing 1 L of MCT oil in a tank and continuously passing it through a custom-made NB generator using a diaphragm pump. During the first 5 min, pure air was injected, and the device was operated to degas other gases. The operation of the generator was stopped when the concentration of NBs reached 2.99 billion/mL. The number of NBs was measured using the NTA device. For consistent measurements, the same condition of laser wavelength, slider shutter, slider gain, and detect threshold was used as 532 nm, 1500, 330 and 5 respectively.

Franz cell experiments: MCE membrane was fitted between the donor and receptor chamber of the Franz cell. Receptor chamber was filled with the ethanol aqueous solution (40% *v/v*). Nile red solution (500 μM, 1 mL) was loaded to the donor chamber. Aliquots (100 μL) were extracted from the receptor chamber and the same volume of ethanol aqueous solution (40% *v/v*) was added after extraction. Collected aliquots were analyzed using microplate reader. 

Biocompatibility analysis of NB oil: Cell studies were performed to evaluate the biocompatibility of the NB Oil using the B16F10 (melanoma cells) cell line. The cells with a density of 3 × 10^3^ cells/well were cultured overnight in DMEM containing 10% FBS and 1% penicillin/streptomycin at 37 °C in a 5% CO_2_ incubator. Thereafter, 100 μL NB oil was added to each well, which was filled with 900 μL DMEM containing 10% FBS, 1% penicillin/streptomycin, and 1% ethanol. After 24 h, the MTT assay was conducted using a microplate reader at 570 nm absorption.

Live and dead cell assay: B16F10 (melanoma cells) cell line with a density of 5 × 10^3^ cells/well was cultured for 24 h in DMEM containing 10% FBS and 1% penicillin/streptomycin at 37 °C in a 5% CO_2_ incubator. Thereafter, NB oil was added to each cell. After 24 h, the live/dead cell assay was conducted using confocal microscopy. The distribution of live/dead cells was visualized by using fluorescein diacetate (FDA) and propidium iodide (PI) staining methods, respectively. 

Depigmentation test: The B16F10 cell line with a density of 1 × 10^3^ cells/well were cultured overnight in DMEM containing 10% FBS, 1% penicillin/streptomycin, at 37 °C in a 5% CO_2_ incubator. Thereafter, α-bisabolol or aliquots were added to the cell and incubated for 4 h. After refreshing the media, alpha-melanocyte stimulating hormone (α-MSH) with known concentration (1 μM) was added and incubated for 96 h. Over time, melanin concentration was measured using the microplate reader.

Depigmenting index calculation: The concentration of melanin measured with a microplate reader is converted to a depigmenting index using the formula below. Melanin (untreated) is the concentration of melanin obtained from B16F10 cells incubated without α-bisabolol, and melanin (treated) is the concentration of melanin obtained from B16F10 cells incubated with an aliquot containing α-bisabolol.
(1)$$\mathrm{Depigmenting}\;\mathrm{index}\;\left(\mathrm{DI},\;\%\right)=\frac{\mathrm{Melanin}\;\left(\mathrm{untreated}\right)-\mathrm{Melanin}\;\left(\mathrm{Treated}\right)}{\mathrm{Melanin}\;\left(\mathrm{Treated}\right)}\times 100\;$$

## 3. Results and Discussion

General bubbles disappear quickly in the solution because of the buoyancy, which is an upward force exerted by a fluid higher than the gravity force of the bubbles, which is a downward force. The low stability of the bubbles in the solution is a major challenge for the application of bubbles, which are non-toxic and can effectively deliver various active ingredients. Therefore, for the bubbles to retain in the liquid, Brownian motion must be greater than their buoyancy to counteract the upward force. Brownian motion is the random motion of particles suspended in a medium and is affected by several factors, including the viscosity of the medium, absolute temperature, and spherical particle size. The Stock–Einstein equation explains that the Brownian motion increases as the particle size decreases at the same temperature and medium. Therefore, the generation of nanometer-sized bubbles is important to retain the bubbles in a liquid for a long time. 

In this study, a previously patented device was used to produce the NB oil. This device constitutes of oil tank, generator, oil pump and gas pump. As oil and air that were stored in the oil tank circulate through the device by the pump, the gas-liquid mixture continuously collides with the structure in the generator to produce the NBs. As shown in Figure 1a, the generated NB oil shows a different phenomenon compared to the pure oil under the green light laser irradiation. NBs dispersed in the oil scatter the light and lead to the Tyndall effect, whereas the pure oil does not show a special phenomenon. NBs were further confirmed by NTA to measure the size and concentration. The enerated NB oil was diluted for accurate analysis because the high density of the NBs leads to interference between the light scattered by the NBs during the measurements. The concentration and size of the NBs were 2.99 billion/mL and 145.9 nm (Figure 1b).

The use of animals in dermal delivery testing in the field of cosmetics has been forbidden in recent years, and various experiments are being replaced with non-animal methods [33]. In particular, the Franz cell experiments are proposed as a reliable alternative to animal testing for skin absorption testing [34,35,36,37,38]. In this experiment, the Franz cell was used to confirm the skin delivery effect of the active ingredients. The Franz cell consists of two compartments (donor and receptor chambers) and a membrane is fitted between the chambers to form a passageway for the active ingredient (Figure 2a). The active ingredient can be delivered to the receptor chamber only through the membrane. Nile red was chosen to observe the active ingredient delivery and the concentration can be measured by fluorescence. The NB oil containing the Nile red was loaded into the donor chamber, and the receptor chamber was filled with aqueous ethanol solution (40% *v/v*), which has a high solubility of Nile red, thus preventing the precipitation. Aliquots (100 μL) were extracted from the receptor chamber to confirm the amount of Nile red that was delivered, and that the same volume of the ethanol aqueous solution (40% *v/v*) was added to maintain the volume of solvent in the receptor chamber. The change in the concentration of Nile red according to the addition of the solvent was corrected through the calculation. Aliquots collected from 0.5, 1, 2, 4, 8, 12, and 24 h at room temperature were analyzed using a microplate reader, and those results were calculated to the concentration (Figure 2b). Strikingly, a higher concentration of Nile red was observed in the presence of NBs than in the absence of NBs. After 24 h, the final concentration of Nile red was 2.06 mM in the absence of NBs and 3.00 mM in the presence of NBs, which was about 1.45 times higher (Table 1). This improved delivery effect is thought to be due to the high Brownian motion of the NBs, which increases the diffusion of the active ingredient and the charge balance effect of the NBs, which improves the movement of the active ingredient in the membrane.

To further demonstrate the biocompatibility of our novel system, in vitro experiments of the pure NB oil and the NB oil containing α-bisabolol with the B16F10 melanoma cell were undertaken. A MTT assay was performed to analyze the cell viability after incubation for 24 h with the NB oil that contained the different concentrations of α-bisabolol. The developed method shows nontoxicity at all concentrations and in the pure oil (Figure 3b). In addition, the B16F10 cells were incubated with the NB oil to observe the cell viability using confocal. After incubation, the cell line was treated with FDA (green) for staining live cells and PI (red) for staining dead cells. As shown in Figure 3a, it was confirmed that most of the cells were live in the presence of the NB oil, which was not significantly different from that in the absence of the NBs. This result suggests that the NB oil has a high biocompatibility due to the use of only the solvent and the active ingredients; this means that it can be a safe system for the dermal delivery of various ingredients. After successfully demonstrating the delivery experiments of Nile red, a delivery test was further conducted using α-bisabolol, which is a representative depigmenting ingredient, to inhibit melanin synthesis related to hyperpigmentation. The B16F10 melanoma cells were treated with aliquots that were collected from the receptor chamber after 12 h, and the melanin concentration was measured using the microplate readers to calculate the depigmenting index [48]. The B16F10 melanoma cells, one of the skin cancers, produce melanin when incubated with α-MSH. However, when α-bisabolol, which inhibits the enzymes involved in melanin production, is applied to the cells, melanin production is reduced. Therefore, the amount of delivered α-bisabolol is estimated by comparing the calculated depigmenting index. The depigmenting index with both aliquots collected in the presence of the NBs shows that the depigmenting index was significantly increased to 400% by the delivered α-bisabolol, whereas a relatively small increase in the depigmenting index to 120% was observed with the aliquots collected in the absence of the NBs (Figure 3c). The difference in this depigmenting index is because of the delivered amount of α-bisabolol, which suggests that the NBs enhance the delivery of the active ingredients without physical stimulation or complex processes.

## 4. Conclusions

Dermal delivery through the skin has been researched with significant interest in the medical and cosmetic fields because of its simple and non-invasive administration. We successfully developed a method that can effectively deliver active ingredients without complicated equipment or procedures using NB oil. The NBs with a concentration of 2.99 billion/mL and a size of 145.9 nm was used for experiments, and when the delivery effect of the active ingredients was compared, it was confirmed that the amount of the delivered ingredients was higher in the presence of NBs than in the absence of NBs regardless of temperature. Moreover, it was confirmed that when an aliquot collected in the presence of the NBs was added to the B16F10 melanoma cells, the depigmenting index increased 3.3 times compare to when the aliquot collected in the absence of the NBs was added. These experimental results indicate that NBs have the potential as a novel technique in the field of dermal delivery.

## Figures and Tables

**Figure 1 nanomaterials-12-03286-f001:**
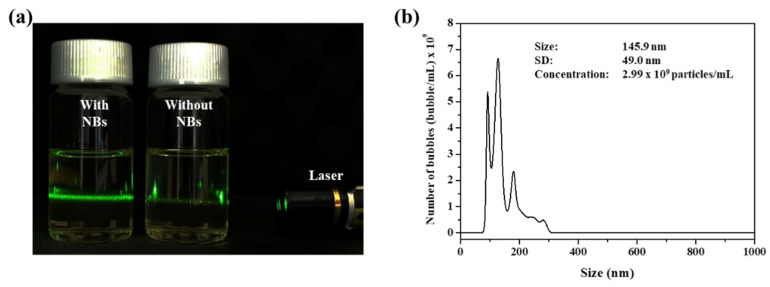
(**a**) Photograph image of NB oil and pure NBs in the presence of green laser irradiation. (**b**) NTA data of generated NB oil.

**Figure 2 nanomaterials-12-03286-f002:**
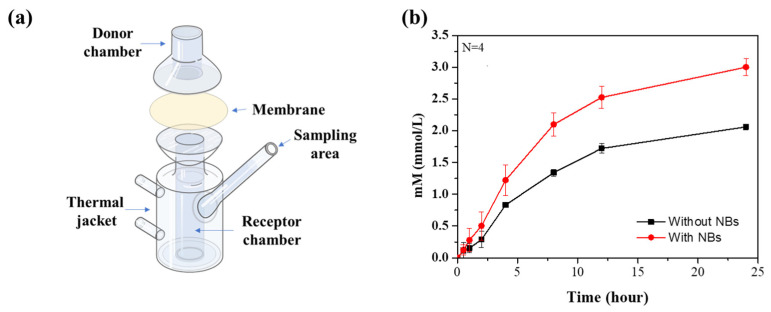
(**a**) Illustration of Franz cell. (**b**) Cumulative concentration of ingredients across the membrane at room temperature.

**Figure 3 nanomaterials-12-03286-f003:**
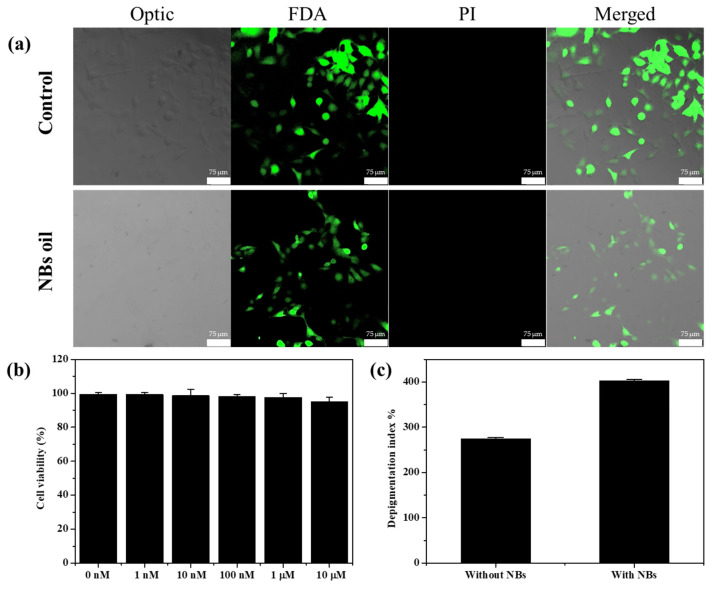
(**a**) Live/dead assay of B16F10 cell line which was stained with FDA (green) PI (red) after being incubated with pure NB oil. (**b**) Cell viability analysis for pure NB oil and α-bisabolol containing NB oil in B16F10 melanoma cell after 24 h incubation. (**c**) Calculated depigmenting index using the collected aliquots during the Franz cell experiment.

**Table 1 nanomaterials-12-03286-t001:** Cumulative concentration of ingredients across the membrane over time.

Time (Hour)	0.5	1	2	4	8	12	24
Concentration with NBs (mM)	0.124	0.279	0.501	1.223	2.099	2.526	3.003
Concentration without NBs (mM)	0.115	0.151	0.289	0.832	1.340	1.723	2.062
Comparison (With NBs/Without NBs)	1.1	1.8	1.7	1.5	1.6	1.5	1.5

## Data Availability

Not applicable.

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
