# Peer review of "Effects of Nanobubbles in Dermal Delivery of Drugs and Cosmetics"

_nanomaterials, 2022, doi:10.3390/nano12193286_

Round 1
Reviewer 1 Report
· Ø The work demonstrates preliminary work on synthesis and loading of NB.
Ø As the authors mentioned, the nanobubbles are very sensitive to the admophore pressure, so the authors still need to do some experiments to verify the stability of the nanobubbles.
Ø In Table.1, the comparison (%) (with NBs/without NBs) is confusing.
Ø The authors need to use an active agent and show the effect of the therapy.

Author Response
Reviewer: 1
Comments to the Author
The review article "Effects of nanobubbles in dermal delivery of drugs and cosmetics" discusses the synthesis of nanobubbles. The authors investigated the loading capacity with Nile red and studied the live and dead B16F10 cells.
Comment:
1)The work demonstrates preliminary work on synthesis and loading of NB.
Answer:
Thank you for the valuable comment.
NB is currently receiving a lot of attention as an active substance vehicle. In particular, the improvement of the delivery effect using the ultrasonic responsivity of NB is important in brain cancer therapy (Nano Letters, 2020, 20, 4512; ACS chemical neuroscience, 2015, 6, 1296). However, the effect of NBs itself to increase the delivery effect without external stimulations has not been reported in the field of dermal delivery. Therefore, we thought that this study will present the potential of NB in ​​the field of dermal delivery. In addition, as in your comments, additional research on the stability of NB is currently in progress for another manuscript that focuses on the NBs stability.
Comment:
2) As the authors mentioned, the nanobubbles are very sensitive to the admophore pressure, so the authors still need to do some experiments to verify the stability of the nanobubbles.
Answer:
Since many papers explain the stability of the nanobubble to atmospheric pressure, it is thought that the NBs in our system will also be stable. In addition, experiments related to the long-term stability of NBs under atmospheric pressure were being conducted, and additional experiments were being conducted on the stability of NBs under various conditions. The results of this will be included in the manuscript of another theme. In addition, the stable properties of nanobubbles are described with reference in the manuscript to help readers understand as follows.
“Second, NBs is highly stable in solution for several months because of stronger Brownian motion than the buoyancy22-24.” (Page 2, line 55-57)
Comment:
3) In Table.1, the comparison (%) (with NBs/without NBs) is confusing.
Answer: Thanks for your notice. The Table is corrected
Comment:
4) The authors need to use an active agent and show the effect of the therapy.
Answer: Thank you for the valuable comment.
We conducted a release test using α-bisabolol, an effective substance that inhibits melanin production, and then treated B16F10 melanoma cells with the elution solution to confirm the melamine production inhibitory effect. In the experimental group with nanobubbles, a more effective reduction in melanin production was confirmed, and this information was described in the manuscript as follows.
“After successfully demonstrating the delivery experiments of Nile red, a delivery test was further conducted using α-bisabolol, which is a representative depigmenting ingredient by inhibiting melanin synthesis related to hyperpigmentation. B16F10 melanoma cells were treated with aliquots that were collected from the receptor chamber from 12 hours, and melanin concentration was measured using the microplate readers to calculate the depigmenting index44. B16F10 melanoma cells, one of the skin cancers, produce melanin when incubated with α-MSH. However, when α-bisabolol, which inhibit the enzymes in-volved in melanin production, is treated to the cells, melanin production is reduced. Therefore, the amount of delivered α-bisabolol is estimated by comparing the calculated depigmenting index. The depigmenting index with both aliquots collected in the presence of NBs show that significantly increased the depigmenting index to 400% by delivered α-bisabolol whereas a relatively small increase in the depigmenting index to 120% was observed with the aliquots collected in the absence of NBs (Fig. 3c). The difference in this depigmenting index is because of the delivered amount of α-bisabolol which suggests that NBs enhances the delivery of active ingredients without physical stimulation, or complex process.” (Page, line 209-224)

Reviewer 2 Report
In this manuscript, the authors attempted to investigate effects of nanobubbles in dermal delivery of drugs and cosmetics, and reported some simple results. In fact, based on the results, it is very difficult to extract some scientific sense for the potential readers. Therefore, the authors should extend their investigation to show exact effects of nanobubbles as well as some features of nanobubbles related to the possible performance in dermal delivery of drugs and cosmetics.
Special comments:
(1) In Introduction, the authors claimed "its effects have not been reported yet on the dermal delivery system", which effects were unclear. Also, the novelties are not well known.
(2) How to produce and control these effects, which features such as size, and others are related to the potential applications in the dermal delivery system.
(3) The authors showed some results, which are failed to explain the reasonable reason, for example, Fig. 2b
(4) The similar investigation should be listed to compare each other, which does a favor for the readers to understand the importance.
(5) Reference style should be uniform, for example, refs, 1,4, and 35.
Author Response
Reviewer 2
Comments to the Author
In this manuscript, the authors attempted to investigate effects of nanobubbles in dermal delivery of drugs and cosmetics, and reported some simple results. In fact, based on the results, it is very difficult to extract some scientific sense for the potential readers. Therefore, the authors should extend their investigation to show exact effects of nanobubbles as well as some features of nanobubbles related to the possible performance in dermal delivery of drugs and cosmetics.
Comment:
1) In Introduction, the authors claimed "its effects have not been reported yet on the dermal delivery system", which effects were unclear. Also, the novelties are not well known.
Answer:
Thank you for the valuable comments. The manuscript is corrected as the reviewer suggested.
“Although NBs have attractive advantage in the field of dermal delivery and the cosmetics industry, Effects of NBs on improving the delivery of ingredients without specific stimulus has not been reported yet on the dermal delivery system.” (Page 2, line 64-67)
Comment:
2) How to produce and control these effects, which features such as size, and others are related to the potential applications in the dermal delivery system.
Answer:
Thank you for the valuable comments. A study on the difference in the delivery effect according to the features of nanobubbles will be conducted in the future, and we think that the size and concentration of nanobubbles will affect the delivery effect. The size of NB will affect the Brownian motion, and the concentration of NB is thought to affect total amount of active ingredients, and both properties are thought to affect the delivery effect. In addition, concentration can be controlled by controlling the operating time of the device.
Comment:
3) The authors showed some results, which are failed to explain the reasonable reason, for example, Fig. 2b
Answer:
Thank you for the valuable comments. The manuscript is corrected as the follow.
“This improved delivery effect is thought to be due to the high Brownian motion of the NB, which increases the diffusion of the active ingredient, and the charge balance effect of the NB, which improves the movement of the active ingredient in the membrane.” (Page 4, line 184-187)
Comment:
3) The similar investigation should be listed to compare each other, which does a favor for the readers to understand the importance.
Answer:
Thank you for the valuable comments. The similar investigations related to the improvement of the delivery effect using nanobubbles have been added to the manuscript as follows.
“NBs with these unique characteristics is used for the delivery of the active ingredients in various fields, and have great advantages, especially in the field of the cosmetic industry33-36.” (Page 2, line 60-62)
Comment:
3) Reference style should be uniform, for example, refs, 1,4, and 35..
Answer:
Thanks for your notice about our mistake. The references are corrected as follow.
“1. Kim, B.; Cho, H.-E.; Moon, S. H.; Ahn, H.-J.; Bae, S.; Cho, H.-D.; An, S., Transdermal delivery systems in cosmetics. Biomedical Dermatology 2020, 4, 1-12 (1).” (Page 7, line 253-254)
“4. Ali, S.; Shabbir, M.; Shahid, N., The Structure of Skin and Transdermal Drug Delivery System-A Review. Research Journal of Pharmacy and Technology 2015, 8, 103-109(2).” (Page 7, line 261-262)
“39. Vinardell, M.; Mitjans, M., Alternative Methods to Animal Testing for the Safety Evaluation of Cosmetic Ingredients: An Overview. Cosmetics 2017, 4, 30 (3).” (Page 8, line 328-329)

Round 2
Reviewer 1 Report
The authors updated the article with the previous comments
Reviewer 2 Report
The authors have improved the manuscript.